# Long-lived multilevel coherences and spin-1 dynamics encoded in the rotational states of ultracold molecules

Tom R. Hepworth [1,2,4], Daniel K. Ruttley [1,2,4], Fritz von Gierke [1,2,3], Philip D. Gregory [1,2], Alexander Guttridge [1,2] & Simon L. Cornish [1,2] ✉

Rotational states of ultracold polar molecules possess long lifetimes, microwave-domain coupling, and tunable dipolar interactions. The availability of numerous rotational states has inspired many applications, including simulating quantum magnetism, encoding high-dimensional qudits and generating large synthetic dimensions. However, engineering the coherent superpositions of multiple rotational states needed for these applications is difficult owing to strong differential light shifts. Here, we overcome this challenge using individual molecules confined in near-magic wavelength optical tweezers. Through precision Ramsey spectroscopy, we find the exact magic wavelengths and sensitivities to detuning errors for multiple rotational state superpositions. We find for traps polarised parallel to the quantisation axis, the magic wavelengths are closely clustered enabling long-lived coherence across multiple rotational states simultaneously. Using a generalised Ramsey sequence, we demonstrate second-scale coherent spin-1 dynamics encoded in three rotational states and perform multiparameter estimation. With modest experimental improvements, we predict second-scale coherence across ten rotational states is achievable.

Ultracold polar molecules possess vibrational, rotational, and hyperfine degrees of freedom which form a vast, low-energy, and experimentally accessible Hilbert space. With sufficient control, this complexity offers many avenues for developing quantum technologies[1] and exploring fundamental physics[2]. A particularly attractive degree of freedom is the ladder of rotational states. These states have long radiative lifetimes, are easily coupled with microwave radiation, and support controllable dipolar interactions. Therefore, they can be used to encode models of quantum magnetism[3–6], synthetic dimensions[7–10], and qudits[11–13]. However, for these applications it is crucial to decouple the internal degrees of freedom from external environmental perturbations and noise[14].

To date, rotational states of molecules have been used to study various spin-1/2 systems[15–17] and to encode qubits that can be prepared in maximally entangled Bell states[18–21]. Extending these studies beyond two-level systems is an outstanding challenge, primarily due to the difficulty in realising long coherence times for rotational-state superpositions. Protocols such as dynamical decoupling[22,23] can extend these coherence times for two-level systems, but are not easily generalised to $n$-level systems[24,25].

Decoherence of rotational-state superpositions in molecules is primarily caused by the large differential polarisability between the states of the superposition. As a result, variations in the optical-trap intensity sampled by the molecules lead to rapid dephasing[14,26,27]. Early approaches to minimise the differential polarisability (and resultant dephasing) used trapping light at a magic polarisation[26–29]. We have recently pioneered an alternative approach where the trapping light is at a magic wavelength[30]. This magic-wavelength light eliminates the differential polarisability between two chosen molecular states,

[1]Department of Physics, Durham University, Durham, UK. [2]Joint Quantum Centre Durham-Newcastle, Durham University, Durham, UK. [3]Present address: Institut für Quantenoptik, Leibniz Universität Hannover, Hannover, Germany. [4]These authors contributed equally: Tom R. Hepworth, Daniel K. Ruttley. ✉e-mail: s.l.cornish@durham.ac.uk

enabling second-scale rotational coherence[31] and long-lived entanglement of pairs of individually trapped molecules[21]. In ref. 31, we showed, for the case where the trap polarisation is orthogonal to the quantisation axis, that the exact magic detuning varies substantially depending on the choice of rotational states. This raises the question: can long-lived coherence be achieved for superpositions of multiple rotational states?

In this work, we address this question, seeking to unlock the rotational degree of freedom in molecules for new applications. Specifically, we investigate how near magic-wavelength traps can be used to study systems beyond two levels. We use microwave Ramsey interferometry to perform Hz-level spectroscopy of individual molecules confined in optical tweezers to measure AC Stark shifts of the rotational transitions. From these measurements, we precisely determine the magic wavelength and its sensitivity to changes in laser frequency, intensity, and polarisation for different rotational-state superpositions. Critically, we find that when the polarisation of the trap is parallel to the quantisation axis, the magic conditions for different superpositions are closely clustered in detuning (in contrast to ref. 31). We exploit this to engineer simultaneous second-scale coherence between three rotational levels realising coherent spin-1 dynamics encoded in the rotational states of ultracold molecules. Using a generalised three-level Ramsey sequence, we perform quantum multi-parameter estimation[32,33], demonstrating the utility of the spin-1 coherence. Finally, using the measured rotational-state dependence of the magic-wavelength condition to constrain a theoretical model of the molecular polarisability, we predict that second-scale coherence should be achievable for superpositions involving ten rotational states and discuss the implications of this for near-term applications.

## Results

### Molecular polarisabilities

Generally, the polarisability of a diatomic molecule is anisotropic. Its response to light can be described by components of the molecule-frame polarisability which are parallel ($\alpha_\parallel$) and perpendicular ($\alpha_\perp$) to the internuclear axis. Broadly speaking, if these components are not equal, different molecular states (with differently shaped wavefunctions) experience different polarisabilities and are prone to rapid dephasing[34]. In an idealised picture, such differential light shifts can be eliminated by tuning the molecular polarisability to be isotropic, that is, finding a wavelength where $\alpha_\parallel = \alpha_\perp$. This eliminates tensor light shifts proportional to the anisotropic polarisability, $\alpha^{(2)} = \frac{2}{3}(\alpha_\parallel - \alpha_\perp)$. Such magic wavelengths can be found in the vicinity of an electronic transition that, due to symmetry, only tunes $\alpha_\parallel$[30]. Previously, this approach has been used to realise second-scale coherence between two rotational levels in $^{87}$Rb$^{133}$Cs (hereafter RbCs) molecules[21,31]. However, this idealised picture neglects subtle effects stemming from the rotational structure in the electronic transitions used to tune the polarisability. These effects preclude the existence of a single wavelength that is exactly magic for multiple rotational states simultaneously. Here, we accurately measure these subtle effects, and develop a model that explains their origin. With this new understanding we are able to optimise the wavelength to be nearly magic for multiple rotational states simultaneously.

### Experimental scheme

We study the optimal conditions for simultaneous multilevel coherence by trapping individual RbCs molecules in optical tweezers at wavelength 1145.3 nm[21,35]. The tweezer light is detuned $\Delta \approx +185$ GHz from a nominally-forbidden transition to the ground vibrational level of the b$^3\Pi$ potential (see Methods). This transition has a linewidth of 14.1(3) kHz[36], so the trap is effectively far detuned and loss due to photon scattering on the transition is suppressed. The molecules are initially prepared in the absolute ground state ($N = 0, M_N = 0$), where $N$ is the rotational quantum number and $M_N$ is the projection of the

rotational angular momentum onto the quantisation axis. From this state, we can drive electric-dipole allowed transitions with microwaves up the ladder of rotational states. We focus on the spin-stretched rotational states with $M_N = N$.

### Magic-wavelength spectroscopy

To identify the optimal conditions for magic-wavelength trapping, we use Hz-level Ramsey spectroscopy to measure AC Stark shifts of the rotational transitions. For a given rotational transition, we trap individual molecules in optical tweezers and prepare them in equal superpositions of rotational levels with microwave pulses (see Methods). We allow these superpositions to evolve for time $T$ before mapping the accumulated relative phase (in the rotating frame) onto state populations with a second microwave pulse. We readout the relative state populations by mapping each rotational state to a distinct spatial configuration of atoms[35]. The rate of phase accumulation is equal to the microwave detuning, allowing us to precisely measure the energy of rotational transitions. Unlike previous studies using magic-wavelength traps[31], we strongly suppress molecular interactions by holding molecules in widely spaced ( ~ 4.2 μm) arrays in tweezers with slightly different detunings (see Methods).

Figure 1a shows examples of such Ramsey measurements. Here, we prepare molecules in a superposition of (0, 0) and (1, 1) in tweezers that are polarised parallel to the quantisation axis ($\beta = 0°$). Oscillations in the relative population $P_0$ of the state (0, 0) occur at the detuning of the microwaves from the transition (0, 0) → (1, 1). The three panels are for different tweezer detunings, indicated by the grey vertical lines in Fig. 1b. The small differential light shifts enable long free-evolution times without significant decoherence, allowing us to resolve transition frequencies with Hz-level precision.

To find the magic wavelength for a transition, we repeat the Ramsey measurements for molecules trapped in tweezers of different peak intensities $I$ and detunings $\Delta$. Figure 1b shows the results of these measurements for the transition (0, 0) ↔ (1, 1) when $\beta = 0°$. We fit the data with the general expression (dashed lines)

$$f(I, \Delta) = f_0 + k(\Delta - \Delta_{\text{magic}})I + k'(\Delta - \Delta_{\text{iso}})^2 I^2, \qquad (1)$$

where $f_0$ is the free-space transition frequency, $k$ is a sensitivity constant and the final term allows for the existence of hyperpolarisability. Here $k'$ is a hyperpolarisability constant and $\Delta_{\text{iso}}$ is the detuning where the second-order polarisability vanishes. For the data in Fig. 1b we find no evidence of hyperpolarisability. As a consequence, at the magic detuning $\Delta_{\text{magic}}$, the first-order light shift of the transition is eliminated and $f = f_0$, independent of $I$. This is highlighted in the left panel of Fig. 1c where we replot the same measurements as a function of $I$. Here, the fitted grey dashed lines highlight points with the same $\Delta$ to demonstrate the linear relationship between $f$ and $I$ and the points at $\Delta_{\text{magic}}$ correspond to the horizontal line. From the fit to the measurements in Fig. 1b, we extract a free-space transition frequency $f_0 = 980,385,597.3(2)$ Hz, a magic detuning $\Delta_{\text{magic}} = 185.2980(7)$ GHz, and a sensitivity constant $k = 98(3)$ mHz MHz$^{-1}$ (kW cm$^{-2}$)$^{-1}$. This technique allows us to measure $\Delta_{\text{magic}}$ for this transition with a precision that is more than an order of magnitude greater than previous techniques based on measuring the contrast of the Ramsey fringes[31].

Finally, we note that the absence of hyperpolarisability is a key advantage of using $\beta = 0°$. To highlight this, the right panel of Fig. 1c shows measurements taken around the corresponding magic detuning for the case where $\beta = 90°$. The results are markedly different. We now observe significant hyperpolarisability that results in a quadratic dependence of $f$ on $I$. Crucially, this makes the detuning that nulls the differential polarisability dependent on the peak trap intensity and hence not truly magic. Such hyperpolarisability effects originate from

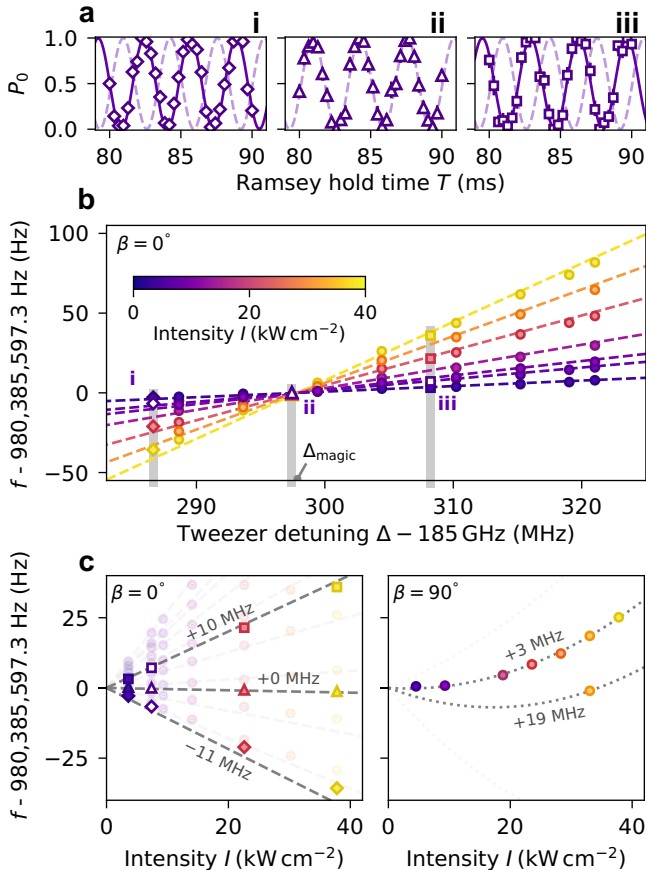

**Fig. 1 | Identification of the magic condition for the transition $(0, 0) \rightarrow (1, 1)$.**
**a** Typical Ramsey fringes for three different values of the tweezer detuning $\Delta$, indicated in (**b**) by the vertical grey lines. Dashed and solid lines are the fit of the Ramsey fringes. The dashed fit line of the trap in (ii) is superimposed on the other plots to highlight the phase slip due to the different transition frequencies. On average, we use 63 experimental shots per data point. **b** Fitted transition frequencies $f$ as a function of tweezer detuning $\Delta$ for many trap intensities $I$. The dashed lines show a fit to Eq. (1). **c** The left panel displays the same data but plotted against $I$ for many $\Delta$ to demonstrate linearity. Again, the dashed lines show a fit to Eq. (1). We highlight three of these lines and label them with the values of $\Delta - \Delta_{\text{magic}}$; these correspond to the grey shaded regions of (**b**). The hollow points correspond to the data fitted from (**a**). The right panel shows similar measurements taken around the corresponding magic detuning for the case where the tweezer polarisation is $\beta = 90°$. Again, the dotted grey lines are a fit to Eq. (1), but now show the existence of hyperpolarisability. Error bars in all panels are $1\sigma$ confidence intervals.

off-diagonal elements in the tensor-polarisability operator that exist for all non-zero $\beta$[34].

## Polarisation and rotational-state dependence of the magic wavelength

The polarisation and rotational-state dependence of the magic-wavelength condition can be understood by decomposing the molecular polarisability into scalar and tensor components. Molecules with $N = 0$ are spherically symmetric and therefore experience only scalar light shifts. However, for molecules that are rotationally excited, this symmetry is broken and they experience both scalar and tensor light shifts. We reformulate the theory of Guan et al.[30] (see Methods) to obtain the total polarisability for the undressed stretched states (with $|M_N| = N$) as

$$\alpha_N(\Delta, \beta) = \tilde{\alpha}_N^{(0)}(\Delta) + \tilde{\alpha}_N^{(2)}(\Delta) C_N P_2(\cos\beta), \quad (2)$$

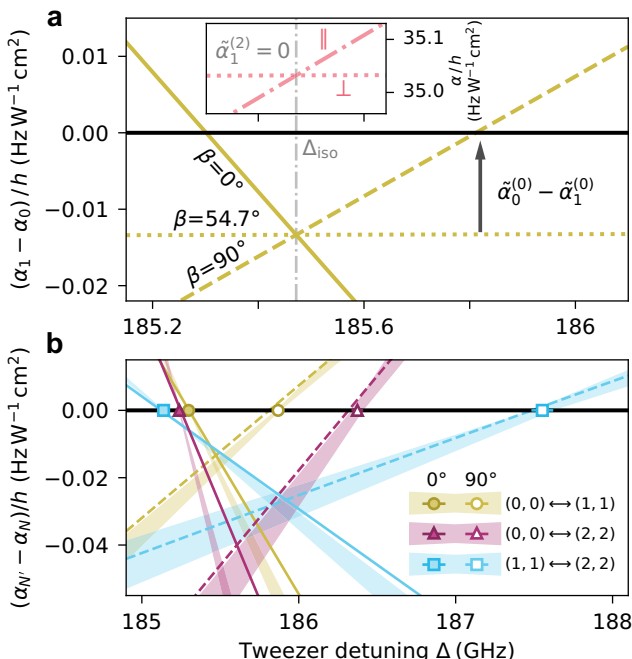

**Fig. 2 | Differential polarisabilities near the magic trapping conditions. a** The differential polarisability $(\alpha_1 - \alpha_0)$ for various laser polarisations. The grey vertical line indicates where $\tilde{\alpha}_1^{(2)} = 0$; the inset highlights that this occurs when $\alpha_\parallel$ (dashed line) is tuned to be equal to $\alpha_\perp$ (dotted line) for $(1, 1)$. At this detuning, the polarisabilities of $(0, 0)$, and $(1, 1)$ are not equal due to a small non-zero differential scalar polarisability $(\tilde{\alpha}_0^{(0)} - \tilde{\alpha}_1^{(0)})$. **b** Points show the measured magic detuning $\Delta_{\text{magic}}$ for each transition and polarisation with $1\sigma$ error bars. The shaded regions indicate the $1\sigma$ uncertainty region of the measured sensitivity constants $k$. These are extracted from measurements similar to that shown in Fig. 1**b**. The solid and dashed lines show the fits of the data to Eq. (2) for $\beta = 0°$ and $\beta = 90°$, respectively.

where $\tilde{\alpha}_N^{(0)}$ is the scalar (isotropic) polarisability and $\tilde{\alpha}_N^{(2)} C_N P_2(\cos\beta)$ is the tensor polarisability (proportional to the anisotropic polarisability $\tilde{\alpha}_N^{(2)}$). Here, $C_N \equiv -N/(2N + 3)$ and $P_2(x) \equiv (3x^2 - 1)/2$. This form of the polarisability makes it explicitly clear that both the scalar and tensor polarisabilities depend on $N$ due to the anharmonic rotational structure in the ground and electronically excited manifolds.

Critically, it follows from Eq. (2) that the exact magic condition $\alpha_N(\Delta, \beta) = \alpha_{N'}(\Delta, \beta)$ occurs at a detuning that is dependent on both the polarisation $\beta$ and the rotational levels $N$ and $N'$. We illustrate the polarisation dependence in Fig. 2a for the transition $(0, 0) \rightarrow (1, 1)$. The tensor polarisability of the state $(1, 1)$ can be eliminated by either tuning the wavelength to $\Delta_{\text{iso}}$ where $\tilde{\alpha}_1^{(2)} = 0$ (vertical dashed-dotted line and inset) or by setting the polarisation to $\beta \approx 54.7°$ (dotted line) such that $P_2(\cos\beta) = 0$[26,28,29]. However, this does not result in magic-wavelength trapping because there remains a small non-zero differential scalar polarisability $\tilde{\alpha}_1^{(0)} - \tilde{\alpha}_0^{(0)}$. To eliminate the overall differential polarisability, it is therefore necessary to introduce a small tensor light shift. The polarisation angle $\beta$ dictates the detuning at which this compensation is achieved. This can be seen in Fig. 2a where the lines for $\beta = 0°$ and $\beta = 90°$ cross through zero at significantly different detunings. We note that $P_2(\cos\beta)$ has turning points at $\beta = 0°$ and $\beta = 90°$, corresponding to values of 1 and $-1/2$, respectively. These polarisations therefore result in magic detunings closest to either side of $\Delta_{\text{iso}}$.

To explore this experimentally, we repeat the Ramsey measurements shown in Fig. 1 but now to determine the magic detuning for $\beta = 90°$. The results are shown in Fig. 2b, along with further measurements for additional rotational-state superpositions, $(1, 1) \leftrightarrow (2, 2)$ and $(0, 0) \leftrightarrow (2, 2)$, for both $\beta = 0°$ and $\beta = 90°$. For each case, the fitted value of $\Delta_{\text{magic}}$ is shown by the point and the measured value of $k$ is displayed

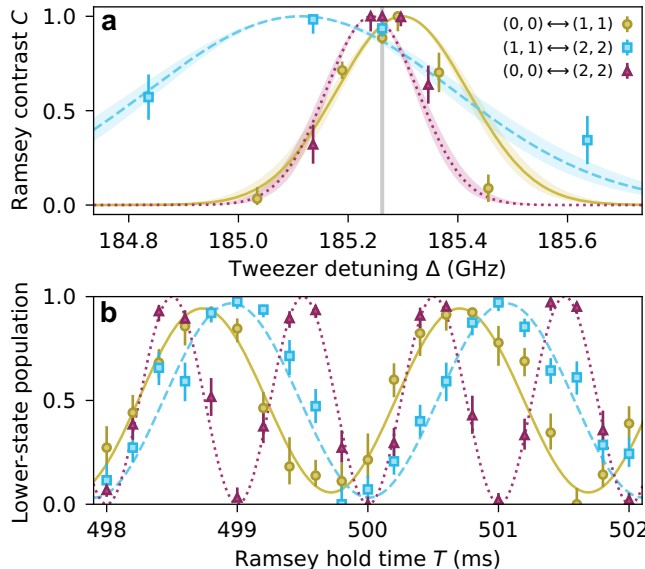

**Fig. 3 | Second-scale coherence on all three rotational-state superpositions.** **a** Ramsey fringe contrast $C$ as a function of tweezer detuning $\Delta$ for each superposition at $\beta = 0°$. The lines are the results of a model fitted to all the measurements with a single free parameter corresponding to the Gaussian intensity noise (see Methods). **b** Ramsey oscillations for the three superpositions at a detuning of ~ 185.26 GHz, indicated by the grey vertical line in (**a**). Error bars in both panels are $1\sigma$ confidence intervals and, on average, we use 18 experimental shots per data point.

as the shaded region. The results clearly show the dependence of the magic detuning on both polarisation and rotational state.

We fit Eq. (2) to the measurements presented in Fig. 2b to constrain two unknown constants related to the molecular structure that are embedded in the terms for $\tilde{\alpha}_N^{(0)}(\Delta)$ and $\tilde{\alpha}_N^{(2)}(\Delta)$ (see Methods). The results are shown by the solid lines for $\beta = 0°$ and the dashed lines for $\beta = 90°$. The agreement between the model and the measurements is excellent. Moreover, having established the parameters in the model, we are able to predict the magic detunings and sensitivities of other rotational-state superpositions (see Discussion).

**Simultaneous second-scale coherence**
Taking each of the rotational-state superpositions studied in Fig. 2b in isolation, we can realise multi-second coherence when tuning the tweezer light to be exactly magic for a given $\beta$. Our molecules are primarily in the motional ground state[35], so the limit to the coherence is mostly from noise on the tweezer intensity and detuning, with a smaller contribution from magnetic-field noise. We expect these to limit the coherence time to $T_2^* \sim 2$ s for the most sensitive superposition, $(0, 0) \leftrightarrow (2, 2)$ with $\beta = 0°$ (see Methods).

Engineering long-lived coherence on all three superpositions simultaneously is more challenging. However, examining the locations of $\Delta_{\text{magic}}$ in Fig. 2b, we see that the magic detunings for $\beta = 0°$ are closely clustered and lie in a window of width ~ 200 MHz. In this region, we can realise robust multilevel coherence. In contrast, for $\beta = 90°$, as used in previous experiments [31], the magic detunings are much further apart and long-lived multilevel coherence is not possible.

To probe the rotational coherence in this region, we use Ramsey interferometry with a hold duration $T \sim 500$ ms. We set the detuning of the microwaves from the one-photon transitions to ~ 100 Hz and use peak intensity $I = 4.6(3)\text{kW/cm}^2$. We measure the contrast $C$ of the Ramsey oscillations as a function of the tweezer detuning $\Delta$ for all three rotational-state superpositions with $\beta = 0°$. The results are shown in Fig. 3a. For all three cases, the observed detuning-dependence of the contrast is consistent with tweezer intensity noise[21] with a standard

deviation of 0.65(4)%, as shown by the lines (see Methods). The different widths of the features reflect the different sensitivities reported in Fig. 2b.

As the tweezer-intensity noise is small, we can engineer simultaneous second-scale coherence for all three superpositions. To do this, we set $\Delta \approx 185.26$ GHz, indicated by the grey line in Fig. 3a. Here, we expect that the $T_2^*$ time for each superposition exceeds 1.5 s (see Methods). In Fig. 3b, we show Ramsey fringes for the three superpositions at a Ramsey hold time $T \sim 500$ ms. We note that the frequency is twice as fast for the superposition of (0, 0) and (2, 2) as both microwave fields used to drive the two-photon transition are detuned. In all cases, we see close to full contrast fringes, demonstrating long-lived coherence for all three rotational-state superpositions at this detuning.

**Spin-1 dynamics and quantum multiparameter estimation**
By operating at the optimum detuning reported in Fig. 3, we can prepare highly coherent quantum superpositions of three rotational states, effectively encoding a spin-1 system in the rotational structure of the molecule. Pushing beyond the usual two-level paradigm will open many new applications in quantum science using ultracold molecules[1]. As a first demonstration of such an application, we use the dynamics of a spin-1 system encoded in the rotational structure to perform quantum multiparameter estimation[32,33]; a technique that has important applications in quantum metrology[37]. Explicitly, we use a generalised Ramsey sequence to precisely measure the relative energies of the three states. We exploit the non-trivial interference of the phases accumulated by the states to produce a complicated interference pattern which is simultaneously sensitive to detunings of both microwave fields from the transition frequencies.

Figure 4a illustrates the generalised Ramsey sequence. First, we use microwave pulses to transfer molecules from the state (0, 0) to an equal superposition of the three states (see Methods). We detune the microwaves from the one-photon transitions $(0, 0) \rightarrow (1, 1)$ and $(1, 1) \rightarrow (2, 2)$ by $\delta_{01} \approx +100$ Hz and $\delta_{12} \approx -150$ Hz, respectively. We allow the superposition to evolve for time $T$ and then perform a sequence of microwave pulses which maps the resulting phases in the superposition onto the populations $P_N$ of all the states (see Methods). To measure the state populations, we extend the multistate readout scheme of ref. 35 to three states. Examples of atomic configurations obtained with this readout scheme are shown in Fig. 4b. In Fig. 4c, we show the state populations after the generalised Ramsey sequence as a function of $T$. The coherence between the three states is seen by the quasi-periodic zero occupation of each state, which is evident even at $T \sim 500$ ms. This simple metric of coherence is directly observable due to our multistate readout scheme; this contrasts to cases where multistate coherence is mapped onto a single measured observable[38,39]. The non-trivial fringes in Fig. 4c are described well by an analytical model, shown by the solid lines, which we use to extract the microwave detunings (see Methods). Using the whole interference pattern, we find $\delta_{01} = 98.11(2)$ Hz and $\delta_{12} = -149.51(2)$ Hz.

The extracted values for the detunings are remarkably precise, despite being measured simultaneously. This is due to quantum interference between the three states that is only possible due to their mutual coherence. We characterise this gain in sensitivity by computing the quantum Fisher information matrix for the equal-superposition state. This quantity, via the Cramér-Rao bound, sets the fundamental limit on the achievable measurement uncertainty of parameters encoded in a quantum state[32,33]. For the equal superposition state, an optimal projective measurement requires only 3/4 of the number of measurements to achieve the same uncertainty on both parameters in this three-level interferometer compared to performing two rounds of two-level Ramsey interferometry (see Methods). Further, the posterior distribution for the two parameters, when measuring in discrete time windows, has far fewer nearby modes of high

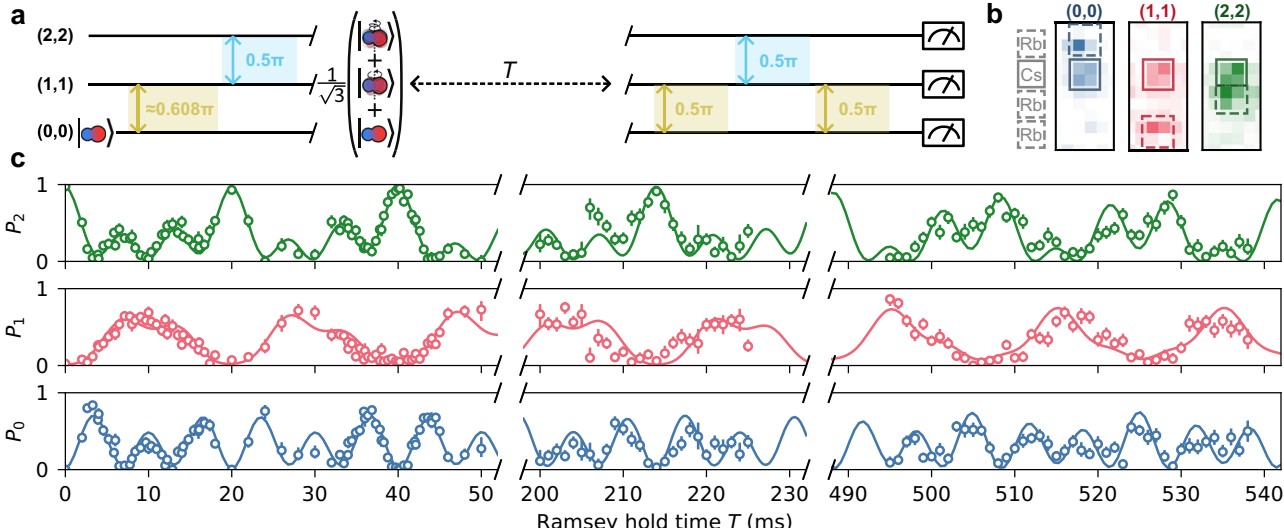

**Fig. 4 | Three-level generalised Ramsey sequence. a** The microwave pulse scheme used in the measurement. The molecules are prepared in the state (0,0). The initial two pulses then form an equal superposition of (0,0), (1,1) and (2,2). The relative phases in the superposition then evolve for a Ramsey hold time, $T$. The final pulses map the phases accumulated to interference in the populations of the states, which are then measured. **b** Averaged fluorescence images showing the spatial configurations of the atoms following molecular dissociation that correspond to each molecular state in the multistate readout procedure. **c** Populations $P_N$ of three states as a function of Ramsey hold time, $T$. Error bars are $1\sigma$ confidence intervals and, on average, we use 38 experimental shots per data point. The whole interference pattern is fitted with an analytic model assuming no decoherence or frequency drifts, shown with the solid lines.

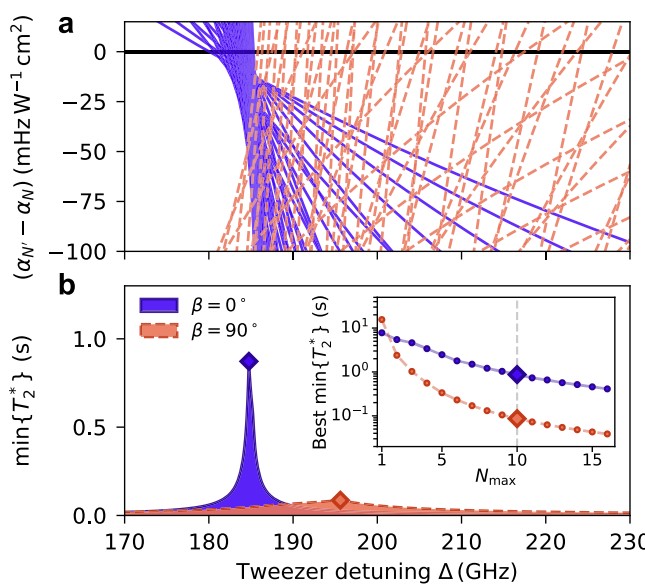

**Fig. 5 | Scalability of the approach for all stretched rotational states up to $N_{max}$.**
**a** Differential polarisabilities for all pairs of states when $N_{max} = 10$, for $\beta = 0°$ (purple solid lines) and $\beta = 90°$ (orange dashed lines). **b** The minimum $T_2^*$ time when $N_{max} = 10$, as a function of tweezer detuning $\Delta$, assuming a relative intensity noise of 0.1% (standard deviation). The inset displays the best min$\{T_2^*\}(\Delta)$ for increasing values of $N_{max}$, the large highlighted points correspond to the peaks extracted from the main plot.

probability than the two-level Ramsey measurement. This is due to the complicated interference pattern and makes the fitting procedure easier. In future, we expect that similar procedures could increase the data-acquisition rate and decrease the uncertainty on measured values when performing spectroscopy of multilevel systems.

These results provide a new perspective in the growing field of multi-parameter quantum sensing and metrology[40], and may inform future tests of fundamental physics using molecules[2,41]. For example, our system is sensitive to changes in two fields (electric and magnetic)

at the same time and crucially can differentiate between them due to the differing differential electric and magnetic moments for each state. This makes molecules interesting candidates for multiparameter quantum sensors[32,42]. Finally, we note that in other platforms, spin-1 systems have already been predicted to outperform their spin-1/2 counterparts as quantum sensors[43,44].

## Discussion
We have demonstrated simultaneous multilevel coherence within the rotational structure of ultracold molecules. We achieve this by individually trapping the molecules in optical tweezers which decouple their rotational states from the environment. Within the rotational manifolds of these isolated molecules, we have encoded spin-1 systems and characterised them with a generalised Ramsey sequence to perform quantum multiparameter estimation.

We predict that these techniques will be generalisable to higher numbers of rotational states, enabling broader exploitation of the rich rotational structure of molecules. For example, in Fig. 5a we show the differential polarisabilities for the 55 possible superpositions when choosing pairs of states $(N, M_N = N)$ from $N = 0$ to $N_{max} = 10$. We give the differential polarisability as a function of tweezer detuning $\Delta$ for $\beta = 0°$ (purple solid lines) and $\beta = 90°$ (orange dashed lines). For $\beta = 0°$, the magic detunings are again closely clustered, enabling simultaneous coherence for all transitions (see Methods). In Fig. 5b, we show the smallest $T_2^*$ among these transitions if the limiting trap-intensity noise were to be modestly improved to 0.1%[45]. This highlights our finding that a trap polarisation $\beta = 0°$ is critical to maximise simultaneous coherence for multiple states: we predict a best multi-state $T_2^*$ of ~0.9 s for the $\beta = 0°$ case, in stark contrast to the limiting $T_2^* \sim 0.1$ s when $\beta = 90°$. In the inset, we show how these multi-state coherence times depend on the maximum rotational quantum number $N_{max}$.

Looking further ahead, our magic-wavelength traps support multi-state coherence for durations much longer than the interaction timescales typical for molecules trapped in optical lattices[15–17] or optical tweezers[18–21]. Therefore, this work paves the way for studies of many-body SU($N$) systems[6,46] and interacting synthetic dimensions[7–10]. For example, using three rotational levels in a pinned array of interacting molecules, one could encode a system of spin-1/2 hard-core

bosons, allowing for representations of bosonic tunnelling and the study of interesting topological effects[47]. The addition of Floquet drives to such a system could be used to study the dynamics of effective bosonic t-J models[48]. Additionally, the long-lived coherence between the rotational states could be exploited to densely encode quantum information in order to form interacting qudits[11–13,49] or quantum memories in hybrid quantum systems[50–52].

## Methods

### Experimental apparatus

Our experimental apparatus has been extensively described in previous works. Briefly, we prepare individually trapped $^{87}$Rb and $^{133}$Cs (hereafter Rb and Cs respectively) atoms in arrays of species-specific optical tweezers[53,54]. We convert Rb-Cs atom pairs into RbCs molecules in the internal ground state with a combination of magnetic-field ramps and laser pulses[35,52,55]. The majority of formed molecules occupy the motional ground state[35].

To obtain experimental statistics, we repeat experimental sequences multiple times. From these statistics, we calculate the relative state populations and estimate $1\sigma$ binomial confidence intervals using the Jeffreys prior[56–58]. We ignore experimental runs in which the requisite atoms were not loaded or we flag molecule formation as unsuccessful. Then, to readout the molecular states, we map them to a distinct spatial configuration of atoms [see Fig. 4b][35]. With additional postselection, we ignore errors common to all states that manifest as apparent molecule loss. The shot numbers given in the figure captions are the number which satisfy these postselection criteria. The apparent molecule-loss errors primarily result from failure to flag unsuccessful molecule formation or Raman scattering of the tweezer light[35]. The former error is independent of sequence length and, for short hold durations, we recover a molecule in ~45% of experimental shots in which we think one was formed. The latter error causes higher loss in longer experimental routines: the molecule lifetime is 3.7(3) s at tweezer intensity $I = 8$ kW cm$^{-2}$. Crucially, both of these loss mechanisms are independent of rotational state so do not skew the relative state populations.

The optical tweezers which trap the atoms and molecules are formed by focusing light with a high numerical-aperture objective lens which is outside the vacuum chamber in which the experiments take place. The experiments in this work are performed after we transfer the RbCs molecules to an array of magic-wavelength tweezers formed from light at wavelength ~1145.3 nm. We prepare arrays of up to four individually trapped molecules. We note that in a given experimental shot, the probability of forming a molecule in an array site was approximately 20%. Therefore in the majority of experimental shots with a molecule, there was only one molecule. The tweezers in this array have $1/e^2$ waists of 1.76(4) μm and, unless stated otherwise, we use peak intensity $I = 4.6(3)$ kW cm$^{-2}$. We form this array from a common source by deflecting light prior to the objective lens with an acousto-optic modulator (AOM) driven with multiple radio-frequency (RF) tones. Each RF tone causes an additional beam to be diffracted from the AOM. Each beam forms a single tweezer and all tweezers are at slightly different frequencies. This means that molecules in the array are non-resonant and their interactions are negligible. This allows us to measure the effect of four tweezer detunings $\Delta$ at once (e.g. for the measurement shown in Fig. 1(b)) over a range spanning ~30 MHz.

The linear polarisation of the tweezers is set by a polariser prior to the AOM and zero-order half-wave plate after it. The light is subsequently affected by several polarisation-dependent optics: an expansion telescope, three large mirrors, the objective lens, and the glass of the vacuum chamber. Therefore, we expect that the polarisation $\beta$ at the molecules could be slightly different to our desired value. Further, there could be a small polarisation gradient across the array, as observed in similar experiments[59]. For each measured transition, the polarisation remains constant and any resulting error is systematic.

When fitting transition frequency data $f$ to the model of Eq. (1), we fit each tweezer trap separately, and quote the values and errors of the extracted parameters as the mean and standard deviation over the traps. From these fits, we estimate there is a tweezer-to-tweezer polarisation shift of ~1°.

We drive transitions between rotational states in the ground manifold ($X^1\Sigma^+$, $v = 0$) with microwave radiation emitted from a dipole Wi-Fi antenna[35]. Allowed electric-dipole transitions are those that satisfy $|\Delta N| = \pm 1$ and $|\Delta M_N| \leq 1$. To prevent off-resonant driving of undesired hyperfine transitions, we limit the microwave Rabi frequencies to ~10 kHz.

### Two-level Ramsey sequences

For each rotational transition that we study, we perform Ramsey spectroscopy to identify the magic trapping condition. For the one-photon transitions $(0, 0) \rightarrow (1, 1)$ and $(1, 1) \rightarrow (2, 2)$, we initialise the molecules in the lower-energy state and perform a $\pi/2$ pulse on the transition with near-resonant microwaves (detuned by ~300 Hz). We wait for a hold time $T$ before performing another $\pi/2$ pulse. This pulse maps the phase that accumulates between the two states onto the state populations, which oscillate at the detuning of the microwaves from the bare transition [e.g., as in Fig. 1a].

When interrogating the two-photon transition $(0, 0) \rightarrow (2, 2)$, we first initialise the molecules in the state $(0, 0)$. We apply a $\pi/2$ pulse on the transition $(0, 0) \rightarrow (1, 1)$ then a $\pi$ pulse on the transition $(1, 1) \rightarrow (2, 2)$. Both microwave pulses are detuned from the one-photon transitions by ~150 Hz. We wait for a time $T$, then invert the pulse sequence. Here Ramsey fringes occur at the two-photon detuning.

After the two-level Ramsey sequences, the population of the states takes the general form

$$(1 + C\cos(2\pi f T + \phi))/2, \tag{3}$$

which we fit to the data. Here, $C$ is the fringe contrast, $f$ is the frequency of the fringes, and $\phi$ is a phase shift of the fringes, typically fixed to $\phi = \pi$. The sensitivity of the phase of the Ramsey fringe with respect to $f$ scales linearly with $T$. For example, assuming a 10% error in resolving the phase (modulo $2\pi$) to achieve a Hz-level error requires measuring Ramsey fringes out to $T \sim 100$ ms.

We circumvent the need to measure all times out to ~100 ms by measuring blocks of fringes separated in time. However, this method forms a very multimodal posterior distribution for $f$. Most error minimisation solvers fail to find the correct mode, or assign correct probabilities to each mode. For this reason, we use the nested sampling Monte Carlo algorithm MLFriends[60] using the UltraNest package[61] to derive the posterior probability distributions for $f$ and assign confidence intervals. We generally minimise the probability weight assigned to other modes by measuring extra fringe blocks at $T/2$, $T/3$, and $T/5$.

We then fit the data $f(\Delta, I)$ with Eq. (1). We fix $k' = 0$ when $\beta = 0°$ as we do not expect hyperpolarisability. For the measurements taken with $\beta = 90°$, we fix $\Delta_{iso}$ to be 185.47 GHz, 185.53 GHz, 185.60 GHz for the transitions $(0, 0) \rightarrow (1, 1)$, $(1, 1) \rightarrow (2, 2)$, and $(0, 0) \rightarrow (2, 2)$ respectively, when fitting $k'$. This is because we have insufficient data to fit both $\Delta_{iso}$ and $k'$ simultaneously. These values are informed by the measurements taken with $\beta = 0°$ and the results in ref. 31. The results of these fits are provided for each transition and polarisation in Table 1.

The hyperfine-free model of Guan et al.[30] [reformulated in Eq. (2)] defines all tweezer detunings $\Delta$ relative to the electronic transition $(X^1\Sigma^+, v = 0, N = 0) \rightarrow (b^3\Pi_0, v' = 0, N' = 1)$. We follow this notation throughout this work and denote the frequency of this transition as $\nu_0$. Experimentally, we can resolve hyperfine structure, so we measure the frequency of the unambiguously identifiable hyperfine transition $(X^1\Sigma^+, v = 0, N = 1, M_F = 6) \rightarrow (b^3\Pi_0, v' = 0, N' = 0, M' = 5)$ at the 181.699(1) G magnetic field at which we operate[36]. We denote this

**Table 1 | Fitted parameters of the transitions that we experimentally study in this work**

| $\beta$ | Transition | $f_0$ (Hz) | $k$ (mHz MHz$^{-1}$ (kW cm$^{-2}$)$^{-1}$) | $\Delta_{\text{magic}}$ (GHz) | $k'(\Delta_{\text{magic}} - \Delta_{\text{iso}})^2$ (mHz (kW cm$^{-2}$)$^{-2}$) |
|---|---|---|---|---|---|
| 0° | (0, 0) → (1, 1) | 980,385,597.3(2) | 98(3) | 185.2980(7) | – |
| | (1, 1) → (2, 2) | 1,960,706,837.3(2) | 38(2) | 185.142(3) | – |
| | (0, 0) → (2, 2) | 2,941,092,437(3) | 184(11) | 185.239(5) | – |
| 90° | (0, 0) → (1, 1) | 980,385,598.3(6) | − 43(2) | 185.86(2) | 25(2) |
| | (1, 1) → (2, 2) | 1,960,706,837(4) | − 18(3) | 187.54(11) | − 580(50) |
| | (0, 0) → (2, 2) | 2,941,092,440(3) | − 63(4) | 186.36(3) | 26(2) |

To determine these parameters, we fit data such as that shown in Fig. 1b to Eq. (1) as described in the Methods.

transition frequency $\nu^{\text{REF}}$. Then, we relate these as $\nu_0 = \nu^{\text{REF}} + 2B_0 + 2B_{v'}$, where $B_0$ is the rotational constant associated with the $(X^1\Sigma^+, v = 0)$ manifold, and $B_{v'}$ is a fitted effective rotational constant of the $(b^3\Pi_0, v' = 0)$ manifold. The tweezer frequency is referenced relative to $\nu_0$ (to ~ 80 kHz uncertainty) through the modes of an ultra-low expansion cavity, which we use to lock the magic-wavelength laser.

**Three-level Ramsey sequence**

Here, we describe the generalised three-level Ramsey sequence that we use when investigating spin-1 dynamics encoded in the molecular rotational structure. First, we generate an equal superposition of the three states (0, 0), (1, 1), and (2, 2). We initialise the molecules in the state (0, 0) and then perform a $2\arccos(1/\sqrt{3})$-radian pulse on the transition (0, 0) → (1, 1), followed by a $\pi/2$ pulse on the transition (1, 1) → (2, 2). As in the two-level Ramsey procedure, these states accumulate relative phases during the Ramsey hold time $T$. After this hold time, we peform a sequence of $\pi/2$ pulses. First, we drive the transition (0, 0) → (1, 1), then the transition (1, 1) → (2, 2), and finally the transition (0, 0) → (1, 1) again [see Fig. 4a]. This sequence causes nontrivial interference in the populations of the three states.

We derive the interference pattern by analytically propagating a pure state through each of the pulses, assuming there is no decoherence or transition drifts. For the sake of simplicity, here we treat the pulses as ideal and assume that the one-photon detunings $\delta_{01}$, $\delta_{12}$ are much less than the one-photon Rabi frequencies $\Omega_{01}$, $\Omega_{12}$. Immediately after the Ramsey hold time $T$, the molecules are in the state

$$|\psi(\delta_{01}, \delta_{12}, T)\rangle = \frac{1}{\sqrt{3}}(|0\rangle + e^{2\pi i\delta_{01}T}|1\rangle + e^{2\pi i(\delta_{01}+\delta_{12})T}|2\rangle), \quad (4)$$

where $|0\rangle \equiv (0, 0)$, $|1\rangle \equiv (1, 1)$, and $|2\rangle \equiv (2, 2)$. After the final pulse sequence, the populations of the states are

$$P_0 = \frac{1}{12}\left(4 - \cos 2\pi\delta_{01}T + (-2-\sqrt{2})\cos 2\pi\delta_{12}T + (2-\sqrt{2})\cos 2\pi(\delta_{01}+\delta_{12})T\right), \quad (5)$$

$$P_2 = \frac{1}{6}\left(2 + \cos 2\pi\delta_{01}T + \sqrt{2}\cos 2\pi\delta_{12}T + \sqrt{2}\cos 2\pi(\delta_{01}+\delta_{12})T\right), \quad (6)$$

and $P_1 = 1 - P_0 - P_2$, which we use to fit the interference fringes. Again, due to the non-trivial $\chi^2$ surface, we fit using the UltraNest package described in ref. 61.

We characterise the gain in sensitivity from our three-level Ramsey sequence by computing the quantum Fisher information matrix for the state that we prepare during the sequence. Generally, the quantum Fisher information matrix for state $|\psi(\vec{p})\rangle$ parameterised by

$\vec{p} = (p_1, p_2, \ldots)$ is given by

$$\mathcal{F}_{ab} = 4\text{Re}\left(\langle\partial_a\psi|\partial_b\psi\rangle - \langle\partial_a\psi|\psi\rangle\langle\psi|\partial_b\psi\rangle\right), \quad (7)$$

where $a$, $b$ are indices of parameters in $\vec{p}$[33]. The quantum multiparameter Cramér-Rao bound is given by

$$\text{Cov}(\vec{p}) \geq \frac{1}{n}\mathcal{F}^{-1}, \quad (8)$$

where $\text{Cov}(\cdot)$ indicates the covariance matrix of the parameters, and $n$ is the number of experimental repetitions[33].

Between the $\pi/2$ pulses of an ideal two-level Ramsey sequence connecting states $i$ and $j$, the state is given by $|\psi(\delta_{ij}, T)\rangle = (|i\rangle + e^{2\pi i\delta_{ij}T}|j\rangle)/\sqrt{2}$. The quantum Fisher information of $\delta_{ij}$ for this state is then given by $\mathcal{F} = (2\pi T)^2$, which implies $\text{Var}(\delta_{ij}) \geq 1/(n(2\pi T)^2)$. For the state $|\psi(\delta_{01}, \delta_{12}, T)\rangle$ that we prepare with the three-level Ramsey sequence [Eq. (4)], the quantum Fisher information matrix is given by

$$\begin{pmatrix} \frac{8}{9}(2\pi T)^2 & \frac{4}{9}(2\pi T)^2 \\ \frac{4}{9}(2\pi T)^2 & \frac{8}{9}(2\pi T)^2 \end{pmatrix} \quad (9)$$

Inverting this matrix, the Cramér-Rao bound states $\text{Var}(\delta_{01})$, $\text{Var}(\delta_{12}) \geq (3/2)/(n(2\pi T)^2)$. Therefore, we would require $\text{Var}(\delta_{12})/(2\text{Var}(\delta_{ij})) = (3/2)/2 = 3/4$ times as many measurements in the three-level case to achieve the same variance bound as two individual two-level Ramsey sequences for a projective measurement that saturates the Cramér-Rao bound. Note that the optimal projective measurement can be dependent on the parameters one is trying to measure (as is the case in general for multiparameter estimation problems), and requires complicated adaptive measurement protocols to continually saturate the Cramér-Rao bound[62]. Removing the final $\pi/2$ pulse on the transition (0, 0) → (1, 1), shown in Fig. 4 produces an interferometer that at intermittent times $T$, fully saturates the Cramér-Rao bound. This however, would be at the expense of not directly seeing preservation of all coherences in all state populations.

**Limitations to two-state coherence**

The coherence time for a rotational transition is limited by noise $\sigma$ on the transition frequency (standard deviation). We consider this to be shot-to-shot noise, such that, for a Ramsey measurement with hold time $T$, there is Gaussian decay in the contrast $C(T) = e^{-(T/T_2^*)^2}$, where $T_2^* = \sqrt{2}/(2\pi\sigma)$ is the coherence time[63].

The primary cause of $\sigma$ is variation in the differential light shift for a given transition. Near to the magic condition, when $\beta = 0°$, the differential light shift is $kI(\Delta - \Delta_{\text{magic}})$. Therefore, noise in the tweezer intensity ($\sigma_I$) or detuning ($\sigma_\Delta$) can map to noise on the transition frequency. Ex-situ, we have characterised the magic-wavelength tweezers and measured relative intensity noise $\sigma_I/I$ ~ 0.7% and placed an upper bound on the frequency noise $\sigma_\Delta \lesssim 80(20)$ kHz with a beat-note measurement.

**Table 2 | Sensitivities of rotational-transition frequencies in the ground manifold ($X^1\Sigma^+$, $v$ = 0) to external magnetic and electric fields**

| Transition | Magnetic-field sensitivity (Hz G$^{-1}$) | Electric-field sensitivity (mHz (mV cm$^{-1}$)$^{-1}$) |
|---|---|---|
| (0, 0) → (1, 1) | 4.73 | −10.9 |
| (1, 1) → (2, 2) | 4.73 | −2.44 |
| (0, 0) → (2, 2) | 9.45 | −13.3 |

The sensitivities are calculated with Diatomic-Py[68]. We calculate the electric-field sensitivities assuming a bias field of 60 mV cm$^{-1}$.

**Table 3 | Values and sources of the parameters used when fitting Eq. (2)**

| Parameter | Value | Source |
|---|---|---|
| $B_0$ | 490.173 994(45) MHz | 69 |
| $B_{v'}$ | 518.0(4) MHz | This work |
| $\alpha_\perp^{\mathrm{bkgd}}$ | 35.3(8) Hz (W cm$^{-2}$)$^{-1}$ | This work |
| $\alpha_\parallel^{\mathrm{bkgd}}$ | 134.4(8) Hz (W cm$^{-2}$)$^{-1}$ | This work |
| $\omega_{v'=0}$ | 261.569 87(6) THz | 36 |
| $\Gamma_{v'=0}$ | 14.1(3) kHz | 36 |
| $\omega_{v'=1}$ | $\omega_{v'=0}$ + 1493.782274(2) GHz | 36 |
| $\Gamma_{v'=1}$ | 8.1(3) kHz | 36 |
| $\omega_{v'=2}$ | $\omega_{v'=0}$ + 2983.743109(2) GHz | 36 |
| $\Gamma_{v'=2}$ | 1.44 kHz | 30 |
| $\omega_{v'=3}$ | $\omega_{v'=0}$ + 4469.88254(2) GHz | 36 |
| $\Gamma_{v'=3}$ | 0.206 kHz | 30 |

The dominant contribution to $\sigma$ depends on the detuning from the magic-trapping condition. The two noise sources are independent such that $\sigma = k\sqrt{\sigma_I^2(\Delta - \Delta_{\mathrm{magic}})^2 + I^2\sigma_\Delta^2}$. For conditions further from the magic condition, the tweezer-intensity noise dominates. Then, to a good approximation, $\sigma \approx k\sigma_I(\Delta - \Delta_{\mathrm{magic}})$ and $T_2^*$ can be calculated accordingly. This is true for the data in Fig. 3a that we use to fit $\sigma_I/I = 0.65(5)\%$. In contrast, when closer to the magic condition, the tweezer-detuning noise dominates and $\sigma \approx kI\sigma_\Delta$. The coherence time for the most sensitive transition we study in this work (that is, (0, 0) → (2, 2) at $\beta = 0°$) is bounded to $T_2^* \gtrsim 3$ s when $\Delta = \Delta_{\mathrm{magic}}$, limited by our measured upper bound on the tweezer-detuning noise $\sigma_\Delta \lesssim 80(20)$ kHz. We include the effects of $\sigma_I$ and $\sigma_\Delta$ when calculating the $T_2^*$ values shown in Fig. 5, using $\sigma_\Delta = 80$ kHz, $I = 4.6$ kW/cm$^2$, and $\sigma_I/I = 0.1\%$[45].

When the molecules are trapped in the magic-wavelength tweezers, the next limitation on achievable coherence times is noise on the magnetic and electric fields in our apparatus. Table 2 gives the magnetic- and electric-field sensitivities of the transitions within the ground manifold ($X^1\Sigma^+, v = 0$) that we study in our experiments. We work with stretched rotational states where the Zeeman shifts due to nuclear spins are identical and the differential magnetic moments arise from the very small rotational Zeeman effect. This means that the differential magnetic moments $(N' - N)g_r\mu_N$ are constant with magnetic field. Here, $g_r$ is the rotational g-factor (for RbCs, $g_r = 0.0062$[64]).

The transition that we study experimentally with the largest magnetic sensitivity is the transition (0, 0) → (2, 2). We measure the magnetic-field noise to be ~ 10 mG by driving hyperfine transitions in Rb. The associated noise on the transition frequency adds in quadrature with that from the tweezers, and limits the coherence time to $T_2^* \sim 2$ s. In our experiment, this magnetic-field noise is only significant at the large magnetic fields (~181.7 G) that we use for molecule formation and dissociation[35]. It is much smaller when operating at low

fields (~5 G) and, in future, we plan to switch off the large field before performing experiments that require longer coherence times. Accordingly, we ignore this source of dephasing for the calculations shown in Fig. 5.

The transition (0, 0) → (2, 2) also has the largest electric-field sensitivity, calculated at a bias field of 60 mV cm$^{-1}$ that we have measured with Rydberg spectroscopy of Rb. The electric-field noise in our apparatus is ~ 2 mV cm$^{-1}$ which is sufficiently low that it bounds the coherence time to $T_2^* \sim 10$s.

## Magic-wavelength model

The isotropic and anisotropic polarisabilities in Eq. (2) are a reformulation of the hyperfine-free model of Guan et al.[30]. They take the form

$$\tilde{\alpha}_N^{(0)}(\Delta) \equiv \alpha_{\mathrm{bkgd}}^{(0)} + \alpha_{\mathrm{mod}}^{(0)}(N, \Delta), \text{ and} \quad (10)$$

$$\alpha_N^{(2)}(\Delta) \equiv \alpha_{\mathrm{bkgd}}^{(2)} + \alpha_{\mathrm{mod}}^{(2)}(N, \Delta). \quad (11)$$

Here, the constant background terms $\alpha_{\mathrm{bkgd}}^{(0)} \equiv \frac{1}{3}(\alpha_\parallel^{\mathrm{bkgd}} + \alpha_\perp^{\mathrm{bkgd}})$ and $\alpha_{\mathrm{bkgd}}^{(2)} \equiv \frac{2}{3}(\alpha_\parallel^{\mathrm{bkgd}} - \alpha_\perp^{\mathrm{bkgd}})$ result from far detuned poles in the polarisability and are independent of $\Delta$. The modulation terms $\alpha_{\mathrm{mod}}^{(0)}$ and $\alpha_{\mathrm{mod}}^{(2)}$ arise from the vibrational poles in the b$^3\Pi$ electronic state. They take the form

$$\alpha_{\mathrm{mod}}^{(0)}(N) \equiv \sum_{v'} \frac{\pi c^2 \Gamma_{v'}}{2\omega_{v'}^3(2N+1)} \left( \frac{N}{\Delta_{v'} + L_N} + \frac{N+1}{\Delta_{v'} + R_N} \right), \quad (12)$$

and

$$\alpha_{\mathrm{mod}}^{(2)}(N) \equiv \sum_{v'} \frac{\pi c^2 \Gamma_{v'}}{2\omega_{v'}^3(2N+1)} \left( \frac{2N+3}{\Delta_{v'} + L_N} + \frac{2N-1}{\Delta_{v'} + R_N} \right). \quad (13)$$

Here, the sum is over the the vibrational poles with vibrational quanta $v' \in \{0, 1, 2, 3\}$. $\Gamma_{v'}$ and $\omega_{v'}$ are the linewidth and transition frequency respectively of the given vibrational state. $\Delta_{v'}$ is the detuning of the trap light from the rovibrational pole with rotational quantum number $N' = N + 1$. The left and right branch terms

$$L_N = N(N+1)B_0 - [N(N-1)]B_{v'}, \quad (14)$$

$$R_N = N(N+1)B_0 - [(N+1)(N+2) - 2]B_{v'}, \quad (15)$$

contain the rotational constants $B_0$ and $B_{v'}$ for the vibrational levels in X$^1\Sigma$ and b$^3\Pi$, respectively.

We realise the magic trapping condition condition by tuning $\alpha_\parallel$ whilst $\alpha_\perp$ remains approximately constant (i.e., close to its background value $\alpha_\perp^{\mathrm{bkgd}}$)[31]. Near the magic-trapping condition, the overall lab-frame polarisability $\alpha_N$ is effectively isotropic ($\alpha_\parallel \simeq \alpha_\perp^{\mathrm{bkgd}}$), hence $\alpha_N \simeq \alpha_\perp^{\mathrm{bkgd}}$. We measure $\alpha_\perp^{\mathrm{bkgd}}$ by comparing the polarisability of RbCs in the state (0, 0) to the known value of the polarisability of a Cs atom in the same trap. The polarisability of Cs is $\alpha_{\mathrm{Cs}} = 919(3) \times 4\pi\epsilon_0 a_0^3$ at 1145.3 nm[65]. The polarisability of RbCs compared to Cs is

$$\alpha_{\mathrm{RbCs}} = \alpha_{\mathrm{Cs}} \frac{m_{\mathrm{RbCs}}}{m_{\mathrm{Cs}}} \left( \frac{\omega_{\mathrm{RbCs}}}{\omega_{\mathrm{Cs}}} \right)^2, \quad (16)$$

where $m_i$ is the mass and $\omega_i$ is the trap frequency for species $i = \{\mathrm{RbCs}, \mathrm{Cs}\}$. We measure the trap frequencies with parametric heating[66] and compare them to obtain $\omega_{\mathrm{RbCs}}/\omega_{\mathrm{Cs}} = 0.704(7)$. From this we extract the polarisability $\alpha_{\mathrm{RbCs}} = \alpha_\perp^{\mathrm{bkgd}} = 754(18) \times 4\pi\epsilon_0 a_0^3 = 35.3(8)$ Hz (W cm$^{-2}$)$^{-1}$. This is in reasonable agreement with the theoretical prediction of Guan et al. (34 Hz (W cm$^{-2}$)$^{-1}$)[30].

We fit the measured values of $\Delta_{\mathrm{magic}}$ and $k$ in Table 1 to with Eq. (2). The majority of the parameters in this equation are fixed to the molecular constants in refs. 36,67 and the measured value of $\alpha_{\perp}^{\mathrm{bkgd}}$. The remaining two parameters, which we fit, are the effective rotational constant of the excited state $B_{v'}$ and the background parallel polarisability $\alpha_{\parallel}^{\mathrm{bkgd}}$. A complete list of the model parameters, values, and sources is provided in Table 3.

## Data availability

The data that support the findings of this study are available at https://doi.org/10.15128/r2kk91fk55v.

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

## Acknowledgements

We thank Arpita Das, Albert Li Tao, and Luke M. Fernley for sharing their spectroscopy results and Luis Santos and Ana Maria Rey for useful discussions. We acknowledge support from the UK Engineering and Physical Sciences Research Council (EPSRC) Grants EP/P01058X/1, EP/V047302/1, and EP/W00299X/1, UK Research and Innovation (UKRI) Frontier Research Grant EP/X023354/1, the Royal Society, and Durham University.

## Author contributions

T.R.H. and D.K.R. performed the experiments with support from A.G. and F.v.G.; T.R.H. analysed the data with assistance from P.D.G. and D.K.R.; T.R.H. wrote the original draft of the paper and all authors reviewed and edited it; S.L.C. supervised the work and managed funding acquisition.

## Competing interests

The authors declare no competing interests.
