## [Transparent Peer Review file · Nature Communications]

Long-lived multilevel coherences and spin-1 dynamics encoded in the rotational states of ultracold molecules

Corresponding Author: Mr Tom Hepworth

Version 0:

Reviewer comments:

Reviewer #1

(Remarks to the Author)

This paper reports a significant advancement in the field of ultracold polar molecules by demonstrating second-scale coherence between three rotational states in a magic-wavelength optical tweezer. The authors extend the application of magic wavelength trapping of RbCs molecules from two-level spin-1/2 systems to spin-1, showcasing simultaneous coherence for multiple rotational superpositions. They further utilize this coherence to perform a generalized Ramsey sequence, enabling quantum multiparameter estimation. This technique may find important applications in quantum simulation, computation, and metrology. The paper also provides a detailed investigation into magic-wavelength engineering, including the polarizability model, sensitivity to laser intensity and polarization, and the impact of noise sources on coherence.

This work is a valuable contribution to the field of ultracold polar molecules, paving the way for studies of interacting many-body quantum systems with large internal degrees of freedom, as well as applications in synthetic dimensions, SU(N) systems, and qudits.

I believe that this work should be accepted for publication after some improvements.

1. While the title seems to suggest that the paper primarily focuses on demonstrating spin-1 dynamics, much of the content is dedicated to determining and optimizing the magic wavelength and engineering the trapping conditions. While this is a critical enabler of spin-1 dynamics, the balance of the paper leans more toward trap engineering than spin-1 dynamics. After rereading the paper several times, I still have the feeling that the spin-1 dynamics results, while novel and impactful, are somewhat overshadowed by the detailed discussion of magic-wavelength engineering. Also, the generalized Ramsey sequence and quantum multiparameter estimation are presented but not thoroughly discussed in terms of their implications for quantum science. I suggest some adjustments to be made to highlight the spin-1 part more.

2. For the spin-1 dynamics, it will make the result more decisive if the spectroscopy can be performed for different tweezer distances/dipolar interaction strengths.

3. The discussion of polarizability and magic wavelength conditions is thorough but somewhat redundant. It is also noticed that similar results of the magic wavelength were already discussed in previous publications of the group. I think rearrange these sections could allow more space for discussing the spin-1 dynamics results.

4. Please make it clear early in the main text that this is an experiment in optical tweezers, not just "magic wavelength trap". Related to this matter, while the Methods section is detailed and informative, it could be better integrated with the main text to improve readability.

Reviewer #2

(Remarks to the Author)

The manuscript "Coherent spin-1 dynamics encoded in the rotational states of ultracold molecules" by Tom R. Hepworth, Daniel K. Ruttley, Fritz von Gierke, Philip D. Gregory, Alexander Guttridge, and Simon L. Cornish experimentally demonstrates a long coherent lifetime for RbCs molecules undergoing oscillations among three rotational levels. The

authors use trap lasers with a frequency near the transition to the bottom of the $b^3\Pi$ potential. By fine-tuning the laser frequency, they achieve a nearly magic trapping condition, as proposed in the theoretical work cited in Ref. [30]. The idea is to tune the trapping laser frequency such that the tensor dynamical polarizability vanishes leading to a residual dynamical polarizability that remains the same across all rotational states. In reality, due to the complex hyperfine coupling mechanism, the magic condition is only approximately ideal. Reference [30] also proposes using an external electric field to achieve true magic conditions. In this work, the authors demonstrate that even though the condition is only nearly magic, it is already sufficient to achieve a coherent lifetime on the order of hundreds of milliseconds. This is highly important for quantum simulation and quantum information science using molecules as clearly mentioned by the authors. The data is very clean with a clear physical picture underlying the results. I strongly recommend its publication in Nature Communication without any further revision.

Reviewer #3

(Remarks to the Author)

The manuscript by Hepworth et al. reports the first experimental demonstration of the near-magic trapping of three molecular rotational levels, which were utilized for encoding a pseudo spin-1 state. Typically, molecules trapped in optical traps experience rotational decoherence primarily due to dipolar interactions or differential trapping potentials caused by complex molecular energy structures. By isolating a single molecule in an optical tweezer, they successfully suppress interaction-induced dephasing. By selecting an optical frequency close to a forbidden molecular transition, they achieved a balance between the scalar and tensor light shifts across multiple rotational states. The authors extensively examined the coherence among paired rotational states concerning various laser frequencies and polarizations. Leveraging well-characterized second-scale coherence, they observed a generalized three-level Ramsey interference and improved estimation of detunings in the driving microwave fields. The manuscript is well-written, with clear experimental explanations and thorough theoretical analysis.

However, the methods and techniques used closely resemble earlier publications from the same group (ref 21, ref 31 in their citations). Specifically, the chosen 'magic' trapping condition for triple states in this study is nearly the same as the magic trapping condition for only two rotational levels explored in Ruttley et al., Nature 637, 827 (2025) (ref 21). The detailed polarizability analysis and measurements appear to be a natural extension or appendix of the experiment presented in (ref 21). Additionally, the authors investigated multi-parameter estimation using a novel spin-1 Ramsey interference scheme at the single-particle level. It would be interesting to explore spin-1 dynamics with at least a pair of single molecules. There, I think this work is better suited for publication in a more specialized journal than in Nature Communications.

In addition to the general considerations, I have several minor comments for the authors' consideration:

1. In ref 21 and 31, second-scale coherence was achieved between (0,0)-(1,1) and (0,0)-(2,2) without precise magic detuning frequency optimization. This raises the question of whether similar coherence times could be achieved for three rotational levels using the magic conditions described in these references.
2. On page 4, the authors state: "At our usual trapping intensity $I=4.6(3)$ kW/cm², the most sensitive superposition, (0, 0) and (2, 2), is expected to have a sensitivity to laser-frequency deviations of just ~ 510 mHz/MHz, which, even with our laser-frequency stability of $\sigma\Delta = 80(20)$ kHz [21], gives a $T2^* \approx 4$ s." However, based on the provided data (510 mHz/MHz * $2\sigma\Delta = 81.6$ mHz), the calculated $T2^*$ should be approximately 12 s. This discrepancy should be addressed.
3. The manuscript should clarify whether the tweezer intensity values refer to peak intensity or average intensity.
4. The main text should include important values of the molecular lifetime ($T1$) in the optical tweezer, which should be significantly shorter than its $T2^*$ time.
5. The authors should explain why the recovery probability starts at 0.4-0.5 in Extended Data Figure 1. The number of experimental shots performed in figures should also be explicitly stated.

Version 1:

Reviewer comments:

Reviewer #1

(Remarks to the Author)

The authors made appreciable amount of changes following my comments, including change of the title. Now I tend to agree the acceptance of this work in Nat. Comm.

Cold molecule, especially in tweezers, is still a relative new research direction. Using magic wavelength method to increase the coherence time demonstrated by the group is likely to be an important technique for many cold molecule groups on various molecular species. The fact that they now have a nice way to achieve long coherence time across multiple rotational states simultaneously is especially interesting.

The spin dynamics part is still relatively weak, probably due to the large tweezer spacing which makes the inter-tweezer interaction small. But the fact that its signature could be observed really shows the strength of the magic wavelength method.

Reviewer #3

(Remarks to the Author)

The authors have made substantial revisions to the manuscript, significantly improving its quality. They emphasize the importance of the first observation of long-lived coherence in multiple rotational states of ultracold molecules and extend the discussion of scalability to more rotational levels. Given these improvements, I think it is acceptable for publication in Nature Communications.

Manuscript NCOMMS-25-01130-T Hepworth – response to referee reports

We thank the reviewers for their careful reading of our manuscript and insightful comments. Overall, their assessment of our work is positive.

However, the reviewers have raised concerns regarding the lengthy discussion of molecular polarisability and magic-wavelength trapping and the overlap with our previous work. To address these concerns, we have made major revisions to the manuscript.

Firstly, we have refocused the manuscript to emphasise the key experimental advance presented in this manuscript: **the demonstration that long-lived coherence can be achieved across multiple rotational states in polar molecules simultaneously**. This has never been demonstrated before and opens many new opportunities (e.g. new qudit encodings, interacting systems of higher spins, synthetic dimensions, mapping internal states to different models e.g. t-J). We believe this is an important advance that should be communicated to a broad audience. The spin-1 dynamics was intended to be a demonstration of this new capability, but our findings indicate that the approach is easily extendable to more than three internal states.

Secondly, we have taken care to clearly present the new findings of this work and how they go far beyond our previous work published in Nature Physics **20**, 415 (2024) [where we only studied the case where the trap laser is polarised orthogonal to the quantisation axis and the use of thermal gases restricted the precision of our measurements]. Here, by using single molecules in optical tweezers, we have (i) performed more precise spectroscopy of the magic conditions (ii) determined the sensitivities spectroscopically and crucially (iii) extended the measurements to the case where the laser is polarised parallel to the quantisation axis. Through these measurements, we discovered that the magic detunings are very closely spaced for the **new** parallel polarisation case and, as such, we found a regime where we can achieve long-lived coherence across multiple rotational states. This would not be possible using the orthogonal polarisation of our previous work.

This has involved significant changes to the structure and content of the manuscript. Critically, we have largely removed the lengthy theoretical discussion of magic-wavelength trapping and the old figure 1 as requested, leaving a single paragraph to provide context. This has given us space to emphasise the new results and their importance. We also extended the discussion to consider the scalability of our approach to a higher number of internal states and have introduced a new figure 5 which demonstrates that long-lived coherence on up to 10 rotational states simultaneously should be achievable. Additionally, we have removed technical details to make the key advances more accessible to a wide audience.

Further, we have made significant modifications to the structure of the article to conform to the Nature Communications style guide and improve readability.

To better reflect the content, we have retitled the manuscript “Long-lived multilevel coherences and spin-1 dynamics encoded in the rotational states of ultracold molecules”. The spin-1 dynamics is still a key part of our work, but as noted the implications of our results are broader.

Details of these changes and other minor alterations to address the reviewers' comments are given below.

To collate and summarise our substantial changes in one place, for your convenience:

- The title has been revised: “Long-lived multilevel coherences and spin-1 dynamics encoded in the rotational states of ultracold molecules”. This addition emphasises the broader implications of our work, due to the scalability of the approach to many more rotational levels.
- We have removed figure 1 and much of the associated technical discussion, distilling the trap engineering narrative to its core details for the general audience of Nature Communications. The figure's core message has moved into figure 2 of the revised manuscript.
- We have clarified the novelty of this manuscript with respect to Refs. [21,31] of the original manuscript. That is primarily, the strong polarisation dependence of the optimum for multilevel coherence. This has included an additional panel in figure 1 of the revised manuscript discussing hyperpolarisability.
- We have strengthened the discussion of the spin-1 dynamics and generalised Ramsey sequence by further discussing the implications for quantum science, aimed at the broader scientific audience of Nature Communications.
- We have added a new figure (5) to the discussion section of the revised manuscript to highlight the scalability of the magic wavelength trap in the optimum configuration to many more rotational levels.
- We have clarified that these experiments are performed with isolated single molecules in optical tweezers, and that all the dynamics are internal dynamics. We have further clarified the role trap lifetime plays on our postselection method.

We believe the manuscript is now suitable for publication in Nature Communications and look forward to further comments from the reviewers.

Yours sincerely,

The authors

Responses to points raised by Reviewer #1

This paper reports a significant advancement in the field of ultracold polar molecules by demonstrating second-scale coherence between three rotational states in a magic-wavelength optical tweezer. The authors extend the application of magic wavelength trapping of RbCs molecules from two-level spin-1/2 systems to spin-1, showcasing simultaneous coherence for multiple rotational superpositions. They further utilize this coherence to perform a generalized Ramsey sequence, enabling quantum multiparameter estimation. This technique may find important applications in quantum simulation, computation, and metrology. The paper also provides a detailed investigation into magic-wavelength engineering, including the polarizability model, sensitivity to laser intensity and polarization, and the impact of noise sources on coherence.

This work is a valuable contribution to the field of ultracold polar molecules, paving the way for studies of interacting many-body quantum systems with large internal degrees of freedom, as well as applications in synthetic dimensions, $SU(N)$ systems, and qudits.

I believe that this work should be accepted for publication after some improvements.

We thank the reviewer for their endorsement of our work. Below we give detailed responses to their suggestions for improvements which we have incorporated into the revised manuscript.

1. While the title seems to suggest that the paper primarily focuses on demonstrating spin-1 dynamics, much of the content is dedicated to determining and optimizing the magic wavelength and engineering the trapping conditions. While this is a critical enabler of spin-1 dynamics, the balance of the paper leans more toward trap engineering than spin-1 dynamics. After reading the paper several times, I still have the feeling that the spin-1 dynamics results, while novel and impactful, are somewhat overshadowed by the detailed discussion of magic-wavelength engineering. Also, the generalized Ramsey sequence and quantum multiparameter estimation are presented but not thoroughly discussed in terms of their implications for quantum science. I suggest some adjustments to be made to highlight the spin-1 part more.

We thank the reviewer for this comment. Upon reflection, we agree that the original title did not reflect the main emphasis of the paper (and as noted above may have caused confusion). We believe the title “Long-lived multilevel coherences and spin-1 dynamics encoded in the rotational states of ultracold molecules” is more suitable for the revised manuscript. This encapsulates both the discussion of magic-trap optimisation, that we have reformulated for a general audience, the spin-1 dynamics that we use as an example application, and also the scalability to more rotational levels.

In the revised manuscript, the spin-1 results still form an important part of the narrative and we believe their results are impactful. To highlight this, we have introduced a section heading on “Spin-1 dynamics and quantum multiparameter estimation” and have rewritten the subsequent text. In addition, we have extended the discussion of the implications of the multiparameter estimation to appeal to the broader readership of Nature Communications. As part of this, we have added the paragraph:

“These results provide a new perspective in the growing field of multi-parameter quantum sensing and metrology [40], and may inform future tests of fundamental physics using molecules [2, 41]. For example, our system is sensitive to changes in two fields (electric and magnetic) at the same time and crucially can differentiate between them due to the differing differential electric and magnetic moments for each state. This makes molecules interesting candidates for multiparameter quantum sensors [32, 42]. Finally, we note that in other platforms, spin-1 systems have already been predicted to outperform their spin-1/2 counterparts as quantum sensors [43, 44].”

2. For the spin-1 dynamics, it will make the result more decisive if the spectroscopy can be performed for different tweezer distances/dipolar interaction strengths.

We agree with the reviewer that spectroscopy of interacting spin-1 molecules is an exciting future direction for this work.

However, we note that, here, the molecular interactions are **deliberately suppressed** as we wish to focus solely on the internal state control of individual molecules. At our intermolecular spacing ($\sim 4 \mu\text{m}$), the strength of resonant dipolar interactions would be $\sim 1 \text{ Hz}$. This is further suppressed as the molecules across the array are trapped in light at different frequencies (cf. Fig. 1 in the revised manuscript), so they experience different ac Stark shifts making the spin-exchange interaction off-resonant. Moreover, throughout the work we did not control the molecule number loaded into the array; in fact, the majority of the data was acquired with just a single molecule.

To emphasise the non-interacting nature of our experiments, we have added the following paragraph in the revised manuscript:

“Unlike previous studies using magic-wavelength traps [31], we strongly suppress molecular interactions by holding molecules in widely spaced ($\sim 4.2 \mu\text{m}$) arrays in tweezers with slightly different detunings (see Methods).”

We are currently undertaking medium-term technical upgrades to allow us to robustly prepare arrays of molecules in magic tweezers at identical detunings. This will allow us to perform future experiments like those suggested by the reviewer, who we thank for their suggestion. Critically, the current manuscript lays the foundation for such studies.

3. The discussion of polarizability and magic wavelength conditions is thorough but somewhat redundant. It is also noticed that similar results of the magic wavelength were already discussed in previous publications of the group. I think rearrange these sections could allow more space for discussing the spin-1 dynamics results.

We thank the reviewer for this comment and have followed their suggestion.

To make the manuscript more suitable for the broad readership of Nature Communications, we have removed much of the technical detail and theoretical discussion about magic-wavelength engineering. This has included the removal of the original Fig. 1 and the associated lengthy discussion.

We believe that, in the revised manuscript, we have distilled the theory down to the essentials needed to understand the polarisation dependence of the magic trapping conditions, which is critical for understanding the implications of our results. This is a new formulation of the theory of Guan *et al.* [30] which highlights the origin of the rotational state and polarisation dependence of the magic-wavelength condition. This has not been presented in previous publications.

We also reiterate that, whilst we have made similar measurements previously, the new more precise measurements using different polarisations presented here are critical to extending the coherence to multiple levels.

Removal of this redundant material has given us space to discuss the implications of our results further for a general audience. As part of this, we have provided new results in Fig. 5 that predict the performance of the near magic trapping scheme to a higher number of rotational states. This is discussed in the revised manuscript in the paragraph starting “*We predict that these techniques will be generalisable to higher numbers of rotational states...*”. We find it quite remarkable that second-scale coherence should be achievable on 10 rotational states simultaneously.

4. Please make it clear early in the main text that this is an experiment in optical tweezers, not just “magic wavelength trap”.

We regret that this was not clear in the original manuscript. We have emphasised, throughout the revised manuscript, that we trap “*individual molecules in optical tweezers*”, noted clearly in the abstract, introduction, and the results section.

Related to this matter, while the Methods section is detailed and informative, it could be better integrated with the main text to improve readability.

In response to this point and to reflect the Nature Communications style guide, we have integrated the Methods section into the main text of the manuscript. Further, we have

reworded much of the Methods section to improve its flow from the main text. This has improved the readability of the Methods section for a general audience.

Responses to points raised by Reviewer #2

The manuscript “Coherent spin-1 dynamics encoded in the rotational states of ultracold molecules” by Tom R. Hepworth, Daniel K. Ruttley, Fritz von Gierke, Philip D. Gregory, Alexander Guttridge, and Simon L. Cornish experimentally demonstrates a long coherent lifetime for RbCs molecules undergoing oscillations among three rotational levels. The authors use trap lasers with a frequency near the transition to the bottom of the $b^3\Pi$ potential. By fine-tuning the laser frequency, they achieve a nearly magic trapping condition, as proposed in the theoretical work cited in Ref. [30]. The idea is to tune the trapping laser frequency such that the tensor dynamical polarizability vanishes leading to a residual dynamical polarizability that remains the same across all rotational states. In reality, due to the complex hyperfine coupling mechanism, the magic condition is only approximately ideal. Reference [30] also proposes using an external electric field to achieve true magic conditions. In this work, the authors demonstrate that even though the condition is only nearly magic, it is already sufficient to achieve a coherent lifetime on the order of hundreds of milliseconds. This is highly important for quantum simulation and quantum information science using molecules as clearly mentioned by the authors. The data is very clean with a clear physical picture underlying the results. I strongly recommend its publication in Nature Communication without any further revision.

We thank the reviewer for their endorsement of our work.

In response to comments from the other reviewers and the editor, we have made major changes to the structure of the manuscript. This has given us more space to discuss the implications of our work for a general audience. However, throughout these changes, we believe that we have maintained the clear physical picture that underlies our data, whilst making it more suitable for the general readership of Nature Communications.

Responses to points raised by Reviewer #3

The manuscript by Hepworth et al. reports the first experimental demonstration of the near-magic trapping of three molecular rotational levels, which were utilized for encoding a pseudo spin-1 state. Typically, molecules trapped in optical traps experience rotational decoherence primarily due to dipolar interactions or differential trapping potentials caused by complex molecular energy structures. By isolating a single molecule in an optical tweezer, they successfully suppress interaction-induced dephasing. By selecting an optical frequency close to a forbidden molecular transition, they achieved a balance between the scalar and tensor light shifts across multiple rotational states. The authors extensively examined the coherence among paired rotational states concerning various laser frequencies and polarizations. Leveraging well-characterized second-scale coherence, they observed a generalized three-level Ramsey interference and improved estimation of detunings in the driving microwave fields. The manuscript is well-written, with clear experimental explanations and thorough theoretical analysis.

We thank the reviewer for their positive remarks about our work.

However, the methods and techniques used closely resemble earlier publications from the same group (ref 21, ref 31 in their citations). Specifically, the chosen ‘magic’ trapping condition for triple states in this study is nearly the same as the magic trapping condition for only two rotational levels explored in Ruttley et al., Nature 637, 827 (2025) (ref 21). The detailed polarizability analysis and measurements appear to be a natural extension or appendix of the experiment presented in (ref 21). Additionally, the authors investigated multi-parameter estimation using a novel spin-1 Ramsey interference scheme at the single-particle level. It would be interesting to explore spin-1 dynamics with at least a pair of single molecules. There, I think this work is better suited for publication in a more specialized journal than in Nature Communications.

We thank the reviewer for their constructive comments. We regret that the novelty of our work with respect to Refs [21, 31] was not clear in the original manuscript.

Whilst our methods are of course very similar to those used in Ref [21], our measurements and findings are very different. Crucially, we have performed measurements for two different tweezer polarisations that have allowed us to develop a deep and accurate understanding of the subtleties of magic-wavelength trapping in molecules. Such understanding is not needed to optimise the coherence for two levels, as in our previous work. However, it is critical if the coherence is to be extended to multiple rotational states simultaneously – an important challenge if the full potential of ultracold molecules is to be realised experimentally.

We believe this work marks an important advance over previous studies. With our newly presented understanding, we engineer simultaneous and long-lived coherence between many rotational states at the same time – using the spin-1 case as an example. To our knowledge, this is the first time this has been achieved for any molecular system and is a crucial step to unlocking many of the proposed applications of ultracold molecules in quantum science (e.g. new qudit encodings, interacting systems of higher spins, synthetic dimensions, mapping internal states to different models e.g. t-J). We have emphasised this advance in the introduction of the revised manuscript, where we write:

“Critically, we find that when the polarisation of the trap is parallel to the quantisation axis, the magic conditions for different superpositions are closely clustered in detuning (in contrast to Ref. [31]). We exploit this to engineer simultaneous second-scale coherence between three rotational levels realising, for the first time, coherent spin-1 dynamics encoded in the rotational states of ultracold molecules.”

To further emphasise the novelty of this work, and make it more suitable for the wide readership of Nature Communications, we have made major revisions to the structure of the manuscript:

- We have removed Fig. 1 from the original manuscript and most of its associated discussion. Upon reflection, we agree with the reviewer that this was too technical for a wide audience.
- We have used the resultant space to include a discussion of the extension of our results to a higher number of rotational states. This includes the new Fig. 5, which emphasises the new understanding that this work presents: that careful optimisation of trap detuning and polarisation is critical when extending this work to more rotational states. We find it quite remarkable that second-scale coherence should be achievable on 10 rotational states simultaneously!
- Additionally, we have included a discussion of the wider implications of our multi-parameter estimation technique for a broad quantum science audience.
- To reflect these changes and the change in emphasis of the manuscript, we have retitled it “Long-lived multilevel coherences and spin-1 dynamics encoded in the rotational states of ultracold molecules”.

Again, we thank the reviewer for their constructive general comments. Below, we provide point-by-point responses to their minor comments.

In addition to the general considerations, I have several minor comments for the authors' consideration:

1. In ref 21 and 31, second-scale coherence was achieved between (0,0)-(1,1) and (0,0)-(2,2) without precise magic detuning frequency optimization. This raises the question of

whether similar coherence times could be achieved for three rotational levels using the magic conditions described in these references.

Firstly, just to clarify, in both Refs [21] and [31] the detuning of the laser was very carefully optimised to achieve the second-scale 2-level coherence. But this was done somewhat blindly in the sense that we had no understanding to guide the optimisation.

However, the reviewer does raise an important point – the main manuscript did not state clearly the advances made over previous works.

We first characterised the magic-wavelength trapping of RbCs molecules in Ref. [31]. Here, the experiments were done with a bulk gas of molecules, and we used a trap polarisation $\beta=90^\circ$. All these experiments were performed between pairs of rotational levels, and coherences were optimised for these pairs of states. The coherence was sufficiently long (second scale) to study dipolar interactions between pairs of states. However, with this trap polarisation and interactions in the bulk, simultaneous and long-lived coherence for multiple transitions could not be achieved.

In Ref. [21] we implemented similar techniques to Ref. [31], albeit with molecules that are individually trapped in optical tweezers at $\beta=0^\circ$. The focus of Ref. [21] was exploiting a trap tuned to the magic wavelength for the transition (0,0) – (1,1) to use dipolar interactions to entangle pairs of molecules. Following these measurements, we had only nascent understanding that this system could support long-lived multilevel coherence.

The reviewer is correct that the magic conditions in Ref. [21] could support similar coherence times to those presented here. Indeed, these conditions are those at the peak of the yellow curve in Fig. 3a of the revised manuscript. However, crucially, we did not possess or present an understanding of *why* this is the case or *how* we would extend this to more than three states.

In this work, we have aimed to provide a clear and comprehensive account of this understanding to a general audience. We believe that the revised manuscript does this and avoids being weighed down by superfluous technical detail. Furthermore, the addition of Fig. 5 in the Discussion shows how this can be exploited in the future to encode rich a variety of important and interesting quantum problems in the rotational structure of molecules.

2. On page 4, the authors state: "At our usual trapping intensity $I=4.6(3)$ kW/cm², the most sensitive superposition, (0, 0) and (2, 2), is expected to have a sensitivity to laser-frequency deviations of just ~ 510 mHz/MHz, which, even with our laser-frequency stability of $\sigma\Delta = 80(20)$ kHz [21], gives a $T2^ \approx 4$ s." However, based on the provided data (510 mHz/MHz * $2\sigma\Delta = 81.6$ mHz), the calculated $T2^*$ should be approximately 12 s. This discrepancy should be addressed.*

We thank the reviewer for noticing this subtle and regrettable error in the original manuscript. Two errors have compounded in the original manuscript.

Firstly, the frequency sensitivity kI , where $k = 184(11)$ mHz/MHz/(kW/cm²) and $I = 4.6$ (kW/cm²), should equal 850 mHz/MHz. This was correct in Table 1 of the manuscript but the derived quantity was incorrect in the main body of the text. We apologise for this typographical error.

Then, as noted in the Methods, the T_2^* time is given by $T_2^* = \sqrt{2} / (2\pi\sigma)$, where $\sigma = kI\sigma_\Delta = 850 \left(\frac{\text{mHz}}{\text{MHz}}\right) * 80 \text{ kHz} = 68 \text{ mHz}$ is the shot-to-shot transition-frequency noise. This predicts a T_2^* time (set by the tweezer-detuning noise) of 3.3 s. We have updated this in the manuscript, and this error does not affect the main conclusions which we draw.

Additionally, we have now made it clear in the revised manuscript that our measurement of laser-frequency noise is an upper bound derived from a beat-note measurement. Therefore, this value for T_2^* is a lower bound.

However, whilst performing this analysis, we felt that a more detailed discussion on the limits to single-transition coherence would be beneficial to the interested reader. Accordingly, we have added the section “Limitations to two-state coherence” to the Methods. Here, we detail different limits on the coherence, including tweezer-intensity noise, tweezer-detuning noise, magnetic-field noise, and electric-field noise.

In our experiments, the coherence time on the superposition (0,0) and (2,2) is limited to a T_2^* time of ~ 2 s from magnetic-field noise (~ 10 mG) at the high field (181.7 G) that we use for molecule formation. However, for future work, we plan to switch off this large field (removing this source of noise) before performing experiments that require long coherence times. Therefore, this does not present a fundamental limit to multi-state coherence and does not change the conclusions that we draw in the manuscript.

3. The manuscript should clarify whether the tweezer intensity values refer to peak intensity or average intensity.

In response to the reviewer’s comment, throughout the manuscript we have clarified that we quote the peak intensities of the optical tweezers.

4. The main text should include important values of the molecular lifetime (T_1) in the optical tweezer, which should be significantly shorter than its T_2^ time.*

We agree with the reviewer that the manuscript should include a discussion of the dominant losses and molecular lifetimes in our system. However, the molecular lifetime

does not have a significant effect on single- or multi-transition coherences, for reasons we explain here.

The dominant source of molecule loss in our experiment is caused by Raman scattering of the tweezer light. This scattering distributes molecules to different internal states, of which there are a vast number. The sheer number of these states means that we are exceedingly unlikely to scatter into a state that we choose to readout. Therefore, the scattered molecules are dark to our readout procedure due to the state specificity of molecule dissociation. Crucially, this Raman scattering rate does not depend strongly on the rotational state, so this effect does not cause any change in the relative state populations.

We note that this Raman scattering time is not a T1 time in the usual sense. It does not cause spin relaxation between different states in our readout basis.

Typically, to our knowledge, the T1 time is that for spin relaxation from e.g. the state (1,1) to the state (0,0). For RbCs, due to the ~GHz transition frequencies between rotational levels, these times exceed 1000 years [Jacob Blackmore, PhD thesis (2020)] and can be neglected. It is precisely this fact that makes molecules such attractive candidates for many applications. However, these applications require long coherence between many rotational states, which is what we have studied in this work.

For the purposes of this work, Raman scattering merely reduces the proportion of molecules that we recover and readout at the end of an experimental routine. As we postselect all experimental shots on successful molecule readout (to study only the relative state populations), to overcome this effect we need only to take more experimental shots.

For this reason, we believe that this discussion is best seen as a technical detail and have opted to include it in the Methods section rather than the main text. Accordingly, we have added the following sentence to the manuscript in the “Experimental apparatus” subsection:

To obtain experimental statistics, we repeat experimental sequences multiple times. From these statistics, we calculate the relative state populations and estimate 1σ binomial confidence intervals using the Jeffreys prior [52–54]. We ignore experimental runs in which the requisite atoms were not loaded or we flag molecule formation as unsuccessful. Then, to readout the molecular states, we map them to a distinct spatial configuration of atoms [see Fig. 4(b)] [35]. With additional postselection, we ignore errors common to all states that manifest as apparent molecule loss. The shot numbers given in the figure captions are the number which satisfy these postselection criteria. The apparent molecule-loss errors primarily result from failure to flag unsuccessful molecule formation or Raman scattering of the tweezer light [35]. The former error is independent of sequence length and, for short hold durations, we recover a molecule in

~45% of experimental shots in which we think one was formed. The latter error causes higher loss in longer experimental routines: the molecule lifetime is 3.7(3) s at tweezer intensity $I = 8 \text{ kW/cm}^2$. Crucially, both of these loss mechanisms are independent of rotational state so do not skew the relative state populations.

5. The authors should explain why the recovery probability starts at 0.4-0.5 in Extended Data Figure 1.

In response to the reviewer's comment, we have added the following clarifying statement to the methods:

The apparent molecule-loss errors primarily result from failure to flag unsuccessful molecule formation or Raman scattering of the tweezer light [35]. The former error is independent of sequence length and, for short hold durations, we recover a molecule in ~45% of experimental shots in which we think one was formed.

This directs the interested reader to a previous work [Ruttley *et al.*, PRX Quantum **5**, 020333 (2024)] where we thoroughly detail the molecule formation and readout techniques that we use here and studies of the molecular lifetime.

The number of experimental shots performed in figures should also be explicitly stated.

In response to this comment, and to conform to the Nature Communications style guide, we have added the average number of experimental shots to the caption of each figure which contains experimental data.